# Biological Well-Being and Inequality in Canary Islands: Lanzarote (Cohorts 1886–1982)

**DOI:** 10.3390/ijerph182312843

**Published:** 2021-12-06

**Authors:** Begoña Candela-Martínez, José M. Martínez-Carrión, Cándido Román-Cervantes

**Affiliations:** 1Department of Applied Economics, Faculty of Economics and Business, Murcia University, 30100 Murcia, Spain; jcarrion@um.es; 2Department of Management Business and Economic History, Faculty of Economics, University of La Laguna, 38204 San Cristóbal de La Laguna, Spain; croman@ull.edu.es

**Keywords:** nutritional health, inequality of human stature, body mass index, Canary Islands, nutritional transition

## Abstract

Developments in anthropometric history in the Iberian Peninsula have been remarkable in recent decades. In contrast, we barely know about the behavior of insular population groups and infants’ and adults’ growth during the nutritional transition in the Canary Islands. This paper analyzes the height, weight and body mass index of military recruits (conscripts) in a rural municipality from the eastern Canaries during the economic modernization process throughout the 20th century. The case study (municipality of San Bartolomé (SB) in Lanzarote, the island closest to the African continent) uses anthropometric data of military recruits from 1907–2001 (cohorts from 1886 to 1982). The final sample is composed of 1921 recruits’ records that were measured and weighed at the ages of 19–21 years old when adolescent growth had finished. The long-term anthropometric study is carried out using two approaches: a malnutrition and growth retardation approach and an inequality perspective. In the first one, we use the methodology recommended by the World Health Organization (WHO) that is based on z-scores. In the second one, we implement several inequality dimensions such as the coefficient of variation (CV), percentiles and an analysis for height and BMI evolution by five socioeconomic categories. The data suggest that improvements in biological well-being were due to advances in nutrition since the 1960s. They show that infant nutrition is sensitively associated with economic growth and demographic and epidemiological changes.

## 1. Introduction

In recent times, anthropometric measures have been used as indicators to evaluate the net nutritional status and explore inequality in historical populations [1,2,3,4,5]. Economic historians study the relationship between height and body mass index (BMI) and the different socioeconomic contexts since the Industrial Revolution. Anthropometrics history objective is to explore the changes in life standards and the impact of modernization processes and economic growth on nutritional health, human well-being and inequality [6,7,8,9,10,11].

There are two key anthropometric measures of malnutrition: height for age and weight for age. While low height for age indicates “stunting”—a retardation of normal growth—weight for age can be an indicator for under- and over-nutrition [12,13,14]. Both indicators are associated with insufficient nutrition, infectious diseases, environmental aggression and excessive labor exertion. Stunting is related to health and nutrition during infancy, can become permanent and have an impact on mid-life health, increasing the risk of chronic diseases and mortality. In the case of weight, there exists a complex problem called the double burden of malnutrition (DBM) that occurs when one part of the population faces undernutrition while another suffers from over-nutrition [15].

Anthropometric data represent human body influences over time and may provide insights into health inequalities that are not discernible by other indicators [16,17,18,19,20,21]. For past populations, anthropometric data are a valuable source of information about living standards and health inequalities. The absence of data to study the evolution of well-being reaches a major dimension mainly in rural and peasant townships. In many places around the world, until the mid-20th century, the peasantry remained outside the market economy and was located in informal economic scenarios where salaries were barely spread or the salaried workers were not the majority. As a consequence, the economic well-being indicators in historical periods are scarce or non-existent. Thus, anthropometric data from military recruitments from the past are appreciated by historians, development specialists and physical anthropologists.

Military sources have informed about the socioeconomic conditions of male heights since the 19th century and allow the construction of alternative and/or complementary indicators, such as per capita rent and income or real salaries, that are indicators commonly used to measure economic well-being [22,23,24,25]. In addition, weight and body mass index data that are incorporated throughout the 20th century, together with height, allow the exploration of changes in nutrition in the long run and health in rural societies, including small peasant communities [21,26,27,28]. In this case, we are studying the easternmost Canary Island. We analyze the historical anthropometrics of the rural population of SB, a municipality located at the center of the island of Lanzarote (Figure 1), in the province of Las Palmas of Great Canary. The municipality was composed mainly of a peasant population 77 miles from the African coast.

In past decades, anthropometric historical developments in peninsular Spain have been relevant. Studies arising from the biological anthropology and the economic history have revealed the variability in Spanish height, according to environmental and socioeconomic contexts throughout the last 150 years. Climate and ecological conditions, the diet and food consumption habits, socioeconomic, family and cultural environments, apart from genetics, form the determinants of nutritional status and biological well-being. This will explain the significant differences observed among average heights in Spain, a country remarkably characterized by interregional environmental contrasts. Spaniards’ height trends are well known in the long term at a national level, by autonomous community, region and province and also the rural–urban gap and socioeconomic differences [29,30,31,32,33].

Recently, more research into the individual inequality field has been developed. These investigations have been carried out through the use of coefficients of variation (CVs), between classes and social groups, through the HISCO and HISCLASS methodologies of socio professional classification, and through the analysis of social differentiation processes by residential areas inside the towns. All this research has created a wide knowledge about trends and the magnitude of nutritional inequality since the end of the 18th century until the decade of 1980 [33,34,35,36]. Furthermore, worldwide concern about the rapid increase in corporal height in many societies is boosting the number of studies about the origins and trends of obesity in Spain [37,38,39,40,41].

Despite the anthropometrical research achievements in Spain, we barely know about the anthropometrical behavior of Spanish insular population groups and about infant and adult growth during the nutritional transition. We know even less about the anthropometric history of populations from the Canary archipelago during the economic and demographic modernization. The available data of average heights by region or autonomous communities show that inhabitants of the Canary Islands were historically tall, although this geographical area had a relatively low GDP [42].

## 2. The Historical Context and Economic Cycles in Lanzarote

The Canary Islands were one of the less-developed regions during the first stage of the modern process of urbanization. However, men from the Canary Islands were relatively taller than Spaniards in the peninsula, leading us to think about something other than just environmental conditions, in explaining their above-average height (e.g., ethnic origins). Some authors state that genetic factors and ethnical peculiarities, coming from ancient and primitive inhabitants of these islands (*guanches*, of Berber origin) could explain the superior stature of Canary men versus Spaniards. This popular and widely extended belief is based on ancient stories from chroniclers that are collected by Bethencourt, that designate the first nations of Canary islanders as “tall men, robust, strong, with beautiful features”. The heights of those primitive populations, according to archeological studies, normally reach heights above 170 cm [43,44]. However, infant mortality in the Canary Islands was among the highest in Spain, and therefore height cannot be easily related to a low morbidity exposure [32].

Previous anthropometric historical studies have shown that the inhabitants of the archipelago presented better physical well-being standards than the average in Spain during the 19th century. This primacy remained until the end of the 20th century [45,46]. Environmental factors, such as the benignity of the climate without cold winters and with mild temperatures, allowed a more stable agricultural production over the years. A food singularity in the islands, in particular of Lanzarote, is the production and consumption of *gofio*, composed of a non-sifted flour from toasted cereals, usually wheat and corn. This flour was used in different meal preparations. Lanzarote also had a strong presence in the fishing industry that was developed during the 19th century and at the beginning of the 20th. This allowed a richer nutrition that might explain the improvements in biological well-being.

Lanzarote’s economy received an impulse from the middle of the 19th century. The port infrastructure improved with the change of the new capital from Teguise to Arrecife in 1847, the main urban nucleus, and its inclusion among the duty-free and customs-free ports with the Free Ports Law in 1852 [47]. The instauration of free trade increased the exports of cereals, mainly of cochineal (*grana*) to the most important European industries, mostly to Great Britain. The cochineal, used as a purple-colored dye in the textile industries of Europe, was one of the drivers for the economic growth of Lanzarote from 1850 until the 1880s. After the cochineal and cereal crisis, the diffusion of new substitutive cultivations, such as chickpeas, onion, tomato, pumpkin, corn or millet (*millo*) and vines took place, that consolidated in the first half of the 20th century, including the potato [48,49].

The diffusion of the potato occurred very early, from the middle of the 18th century, and especially from the end of the 19th century, like that of corn or millet [50]. The fishing exploitation and the fish canning industry gave an impulse to the local salt industry. However, Lanzarote did not participate in the expansion of the banana economic cycle that was mainly carried out on the two big islands of the archipelago, Tenerife and Las Palmas of Gran Canaria. Lanzarote’s economy was not so dynamic and this explains the importance of emigration to America in the first decades of the 20th century, first to the main island of the Canary Islands and Cuba and afterwards to Venezuela, Uruguay and Brazil. The clandestine emigration lasted until the decade of the 1940s [51,52,53]. Figure 2 shows the demographic crisis in the 1880s and the scarce dynamism of the island’s population and in the city of SB until 1930.

The 1960s marked a change in the economic growth model of the Canary Islands. Lanzarote’s economy experimented a new expansion phase, now driven by the tourism industry. Tourism became the most important sector in the economy. The island of Lanzarote became highly touristic and attracted several millions of tourists each year. Immigration explains the vigorous demographic increase of the last decades of the 20th century. This “tourist miracle” was sustained by improvements in hydraulic infrastructures in the 1960s and the international airport constructed in 1970. Water scarcity has been a challenge for the islands’ inhabitants since ancient times. Runoff water storage was achieved thanks to the construction of *maretas*, hydraulic infrastructures that were improved with the construction of big walls since aboriginal times. The Great Mareta of Teguise had a huge relevance in the main urban core. At the beginning of the 20th century, the water supply was insufficient and, in later decades, galleries, numerous wells, cisterns and above all large deposits (*maretas*) were constructed. Most were supplied with water transported by steamships and tankers from other islands [54].

A final step for drinkable water endowment was the five-year Hydraulic Plan from 1961–1965, that planned the continuation of water tank construction, the collection of groundwater and the carrying out of drilling, reforestation and the effective development of important infrastructures, among which a large desalination plant built in 1965 stood out [55]. The water supply network continued until the end of the 20th century, conditioned by the residential construction development, the services and the push of national and international tourism. Between 1970–1980, agriculture ceased to be decisive although it was transformed by technical innovations, some traditional, such as the *jable* or the *enarenado*, -volcanic sand with which certain crops are covered to conserve the humidity of the soil, avoiding the evaporation of water and thermal differences in the soil.-, that spread and increased the vineyards for wine production. These techniques made it possible to improve the productivity of rainfed soils in Lanzarote. Since the decade of the 1970s, the decline of the powerful fishing fleet (mainly sardine boats), after the decolonization of Western Sahara and its occupation by Morocco, did not impede the maintenance of small artisanal fleets that are still important for fish consumption, a key part of the familiar diet. Livestock also suffered a deterioration, which mainly affected the goats that supplied milk to the traditional cheese industry (majorero cheese). In fact, its demand in the regional Canarian market was reactivated at the end of the 20th century, again promoting the development of the cheese industry [56].

Since [57], height has been used as the main anthropometric proxy for biological well-being, reflecting the nutritional health conditioned by the diet, energy expenditure and labor exertion, among others. Furthermore, anthropometric measures such as height or BMI are an alternative to traditional economic indicators such as the per capita income or real wage indices and offer contrasting results. The main objective of this case study is to identify the evolution of biological well-being from the anthropometric history of SB, a rural municipality located at the center of the island with a large peasant population. In fact, the Monument to Peasantry, by the artist César Manrique, was installed in this locality in 1969 next to the House Museum. We show the relationship between anthropometric changes, mainly in heights, with the nutritional processes and economic modernization. Assuming that adult height is a good indicator of the biological living standards and inequality in the net nutritional status, we focus our attention mainly on the relationships between economic growth and inequality in the evolution of height and its disparity. We verify that the relationships are not identical to BMI dynamics. In addition to analyzing biological well-being, the relationships with social and environmental inequalities and even with life expectancy and infant mortality are studied.

## 3. Materials and Methods

### 3.1. Data Source

The municipality’s selection was determined by the availability and quality of the data and they are very representative Lanzarote’s population. The territory of the municipality of SB has an area of 40.89 km^2^ and represents 4.8% of the island’s area. However, its population represents something else in the island's demography: 9.4% in 1887, 10.11% in 1950 and 13.5% in 2001. The urban nucleus is located in the center of the island, composed mainly of a peasant population until the decade of the 1960s. The territory produces the agricultural cultivations mentioned in the previous section and it has access to the sea, allowing fishing. The township belongs administratively to the province of Las Palmas, with Lanzarote being the easternmost of the Canaries and the closest one to the African coast (Figure 1). Not all the islander populations show the same environmental and socioeconomic characteristics, but they present many similarities. We consider that the municipality sample is a good prototype of the island.

Our anthropometric measures are homogeneous in age between 1906 and 1970 (cohorts from 1885 to 1949) when conscripts were measured at the age of 21 years old. However, the regulation of 1969 set the recruitment age at 20 years old from the following year. Later, between 1970 and 1986 the regulation changed the entry age from to 19 years old. This fact could have affected height and weight, especially the latter. It has been observed that Spanish populations—the majority with healthy food consumption standards—finish their growth at 18 years old [58]. The age changes became stable from 1988 (Figure 3). Given that the growth of the height of the Spanish population is characteristic in modern populations since the 1980s, we do not use height correction techniques nor do we standardize height at age 21 years from this date. We consider that at the age of 19 years the growth of the analyzed populations was completed. The use of age-specific standardization techniques is often done for historical populations that have been under nutritional stress and whose growth lasts until their twenties or twenty-one years of age, even older.

The measurement system also changed and became simpler. The municipalities ceased to carry out their function of enlisting and classifying the recruits, whose control passed to the military recruitment centers (*Caja de Reclutas*) from the 1980s through the presentation of the National Identity Card. This system of *quintas* remained valid until December of 2001. There were also changes in the measurement of heights. Until 1969, height was measured in millimeters, and since then, it has been in centimeters. Equally, the duration of the military service fluctuated in the 20th century. In 1912, military service changed to three years, in 1940, to two years, and in 1986 it was set between fifteen and eighteen months. Finally, in 1991 and in the field of professionalization of the armed forces, the new Law of Military Service shortened the duration to nine months. Universal military service ended on the 31st of December in 2001. In Spain, since the 1st of January 2002, all soldiers, including those of the Royal Navy, are professionals.

We use height, weight and body mass index (BMI) data as biological living standard measures of nutritional health. As in other Spanish anthropometric studies, the data from the analyzed municipality are drawn from the Actas de Clasificación y Declaración de Soldados (ACDS) (Soldiers’ Classification and Declaration Acts) that are preserved in the Quintas Section of the SB Historical Municipal Archive (AMSB). These documents are filed in boxes by recruitment year and have been available at a municipality level in all the Spanish state since the military legislation of 1857. In addition to height and weight, the year of birth, occupations and literacy levels are registered, among other variables. The time series covers the period from 1907 until 2001, corresponding to the cohorts born between 1886 and 1982. The sample comprises all the *quintos* or recruits that showed up for the measurement act or “*talla*” during the period under analysis. In contrast to height, that was measured for all the recruits in the whole period, weight is only reported in later years, for 1912 recruits and from 1955.

In total, we have collected information about 2172 records of conscripts that were called up for military enrollment. The number of observations that report information about height reduces to 1921 cases and 1286 observations report information about weight. Table 1 shows a brief summary of the number of cases for each variable of interest.

As can be seen in Figure 4, our final series are not affected by censored data nor are they truncated. This is important because these used to be some of the main problems of samples that were collected from military records [59]. The frequency distributions of the three main variables of interest in our study follow a quasi-normal distribution (Gaussian) (Figure 4).

Our results consider that height is a measure of adults’ health and also a good informative indicator about economic and epidemiological conditions during childhood. The final adult height reflects the environmental conditions in the first twenty years of life, in which childhood and adolescence are two critical periods for growth. Due to that, the data are presented by birth cohort. In contrast, BMI is an indicator that reports information about the nutritional health status from a certain population at the time of measurement, given that it is conditioned by weight, a variable that changes more frequently from one year to another. In this case, BMI data are presented by year of recruitment. BMI value classification is as follows: low weight (BMI < 18), normal weight (18.5 ≤ BMI < 25), overweight (25 ≤ BMI < 30) and obese (MBI ≥ 30), allowing us to explore consumption habits and healthy routines with respect to nutrition.

### 3.2. Methodology

The methodology we use to study malnutrition is based on z-score computations according to international references used by the WHO (2007) [60] and the national references from Orbegozo [61] and Carrascosa [62]. Next, we explore the evolution of inequality using dispersion measures, examining the weight of malnutrition through those that have short height (stunting), using the coefficient of variation (CV), percentiles compared with standards of modern populations and a socioeconomic analysis of the evolution of anthropometric indicators by educational and social class groups. The coefficient of variation (CV) seems to show inequality better than other dispersion measures [63,64]. In some cases, in order to smooth the trends, we use three-year moving averages.

## 4. Results

Table 2 summarizes the main statistics for the whole sample by ten-year birth cohorts. Body mass index (BMI, kg/m^2^) is only calculated for some decades of the 20th century for which we have weight data available. 

Figure 5 presents the trend of average heights for recruits born between 1886 and 1982. We use 3rd order moving averages in order to smooth the trend given that there are some years that fluctuate a lot because they contain few observations. The series of average heights shows an ascendant trend over time although there are some periods which show a regression. The average height for those conscripts that were born before 1950 was under 170 cm, while for those born from the 1970s onwards we observe an accelerated and more constant increase over time, reaching values above 175 cm. Within those born from 1887–1889 and 1980–1982, the average height went from 167.2 to 176.2 cm, registering an increase of 9 cm.

The periods with greater increases in height are documented at the end of the 1940s (*p*-values < 0.005) and 1960s (*p*-values < 0.005) (*p*-values are calculated from the regression of the variable of interest (height, weight and BMI) as the dependent variable and year of birth/recruitment as dummy independent variables, no other covariates are included). Those of regression among those born at the end of the 19th century (*p*-values > 0.005) and the beginning of the 20th century (*p*-values = 0.000). Among the cohorts from 1887–1889 and 1906–1908,the average height reduced two centimeters, somewhat less if we widen the comparison between 1887–1891 and 1904–1908, with a decrease of 1.3 cm.

Figure 6 shows the weight trend that increases 4.5 kg among the recruits of the five-year groups of 1955–1959 and 1978–1982. We verify that by 1970 there was an accelerated increase caused mainly by conscripts measured in 1970–1974 (*p*-values ≤ 0.05). In 20 years (1955–1974), an increase of 6.2 kg is registered, which we see above all in military recruits at the end of the Spanish “economic miracle” stage. However, during the next decades the average weight seems to stagnate, although in the last years of the 20th century (*p*-values < 0.05) there is a growth rebound that even exceeds the average weight of 68 kg.

Figure 7 presents the average BMI trend by year of birth and recruitment. Average BMI has experienced a continuous decline since the 1970s (*p*-values < 0.05) and stagnated at average values close to 21.5 in the last years of the sample (*p*-values > 0.05). This fact can be attributed to increases in height that are more pronounced with respect to intergenerational weight gains. Furthermore, mean values of BMI for all conscripts analyzed present values that are considered as a healthy weight (18.5–24.99), as is shown in Table 3.

### 4.1. Malnutrition Analysis

In this section, we present z-score evidence for the variable height in order to compare our data with other population standards (Table 4). We use reference values around 18 years old, because we are aware that growth becomes stable from 18 to 20 years old [65]. We have selected the 50th percentile value for the ones registered by the WHO (2007) [60] at 19 years old (height: 176.54 cm; SD: 7.30), the values of Carrascosa et al. [62] for Spaniards at the age of 18 years old (height: 175.97; SD: 6.06) and for adults (height: 177.33; SD: 6.42) and the ones calculated by the Obergozo Foundation [61] that are also for 18-year-old individuals (height: 176.27; SD: 5.69).

Figure 8 presents the four annual HAZ series using different international and national data references. The four HAZ series show an almost identical evolution, independently of the reference considered for their construction. For the whole period analyzed, malnutrition is irrelevant, and it only stands out at the beginning of the 20th century. Although there are not enough observations due to strong emigration, the first decades of the last century are the most vulnerable. Since the 1920s, we observe a remarkable reduction in malnutrition that increases for those cohorts born after 1950–1960.

### 4.2. Inequality Analysis

In this section, we estimate inequality using different measures such as CV, percentiles and the evolution of height and BMI by educational and social class groups. According to some previous studies, it seems that CV expresses inequality better than other dispersion measures [63,64]. A seminal work [63] found a high correlation between the coefficient of variation and differences in average height from different social groups. Based on this evidence, this statistical heterogeneity indicator has been used as an indirect measure of anthropometric inequality and therefore socioeconomic inequality. In this case, we present in Figure 9 the evolution of the CV for heights and BMI for the full sample. The results suggest that height inequality diminished at the beginning of the 20th century, increased in the decade of the 1940s, decreased in the next one and experienced a higher increase from 1960. The BMI inequality analysis suggests that it increased between 1960 and 1980 and at the end of the period.

Figure 10 and Figure 11 show the trend of height and BMI percentiles by decade of birth/recruitment. Height percentiles present a very similar trend for the four percentiles defined (Figure 10). We observe that the 95th percentile shows an upward trend for birth cohorts born after 1970, while the other three percentile categories reflect a downward trend for these birth cohorts. The results in Figure 10 suggest an increase in heights for the tallest at the end of the period while there is a decrease for the rest.

Figure 11 shows the trend for the four BMI percentiles defined. The 25th, 50th and 75th percentiles present a parallel and constant evolution over time. In contrast, the 95th percentile presents a rapid increase in BMI values for birth cohorts born after 1970. There is an increase of more than 5 kg/m^2^.

In order to make comparisons of the evolution of height and BMI, we grouped our data into several social class categories. The social class groups have been defined according to HISLASS classification: students, non-manual workers, manual workers, farmers and farm workers. Table 5 presents the main statistics of height and BMI by social class group for the whole period, given that some groups are not sufficiently represented due to the absence of information in some historical sections. Differences are noticeable and stand out for students, who are the tallest, compared to farmers and farm workers, who show the lowest heights. The height gap between occupations is associated with agrarian activities and the difference of white collar or non-manual workers and students is more than four centimeters. Between the ranks of the tallest and the shortest in stature (students versus farmers and farm workers), the differences reach 6.7 cm. The results do not differ from other studies about the Spanish population in the 20th century, that reveal the biological well-being inequality among occupational groups and social classes, even among neighborhoods in the same city [6,36,64,66,67].

Another inequality dimension can be observed through educational attainment. Figure 12 shows the average height evolution for youths that can only read and write (literate population) compared to students aged 19–21 years, a population group with a higher educational level. In general, we observe an increase in average height for both groups over time. The differences between the two groups were important until the cohorts of the 1940s, and again in the 1960s and 1970s, which favored students. The exception in the 1950s is probably due to the fact that the number of individuals classified in the category of students is low.

Finally, Figure 13 shows the evolution of average height according to the place of origin, that we divide into four groups. Firstly, those born in SB (n = 1539); secondly, those born in Lanzarote but in a municipality different to SB (n = 225); thirdly, those born on a Canary Island different to Lanzarote (n = 55); and, lastly, those born outside the Canaries (n = 42). The data suggest that mobility among islands was present during the whole period and that immigration was noticeable in the last two decades of the 20th century. With height data, we observe an increase over time for all origin groups. Those born in the Canary Islands seem to be taller than those born outside the Canaries for the last two periods. In the period of the “economic miracle” (1950–1960) it stands out that those born in SB are somewhat taller than those from the rest of Lanzarote or the Canaries. Even in the 1970–1982 period, they present an average height somewhat higher than the rest of the individuals from Lanzarote, which indicates the good nutritional status of the population analyzed since 1950.

## 5. Discussion

Preliminary investigations, some of them very early, with data from military records at a regional level had shown that the height of the male inhabitants of the Canary Islands was among the highest in the Spanish regions since the end of the 19th century [29,30,32,42]. More recently, case studies with data on male heights from military records, from recruitments from the municipalities of the province of Santa Cruz de Tenerife (Western Canary Islands) for the cohorts from 1870 to 1915, also confirmed the relative advantage of the male height of island populations versus peninsular Spanish populations. The set of anthropometric history studies suggested that the biological standard of living, an expression coined by Komlos (1993) from adult height, was somewhat higher in the historical populations of the islands than in those of the Iberian Peninsula. It should be noted that, until the middle of the 20th century, the average height of Spaniards was among the lowest of Western European people [1,42,68].

On the other hand, preliminary archaeological investigations carried out a long time ago with skeleton data from the ancient and primitive inhabitants of these islands (*guanches*) suggested that the nutritional status was also relatively favorable, since male heights easily reached 170 cm (43–44). However, the results obtained by physical anthropologists, in addition to referring to populations from more than four centuries ago, are very scarce and fragmentary. In any case, the investigations carried out to date from different fields pointed to the superiority of the male heights of the Canary Islands with respect to the average of those of the Spanish population. Environmental factors, such as climate and access to nutrients, are variables that explain the relative improvement in height and the biological standard of living of the populations analyzed.

This paper yields important results that improve the knowledge of the anthropometric history of the Canary Islands for several reasons. It is the first study to address the evolution of height at the end of adolescent growth in the eastern Canary Islands during the nutritional transition and modernization. The 20th century witnessed dramatic changes in the economy and demography that affected the biological standard of living. With data from the municipality of SB on the island of Lanzarote, the easternmost island of the Canary Islands and close to the African continent, it addresses secular change in a transcendental period of Spanish economic and social history, from the end of the 19th century to the beginning of the 21st century. To do this, we use the male heights of the cohorts from the 1880s to the early 1980s. In addition, unlike previous studies that were more focused on male height, this study analyzes the body mass index with regularly available weight data since 1955. Thus, we discuss the effect of the nutritional transition in individuals and the effects caused by the improvements of economic well-being and of the diet on physical or biological well-being. This is important since we analyze a very broad historical period, at least with height data, which in theory extends from a context of malnutrition or deficient nutrition due to scarcity (late 19th century) to another of satisfactory nutritional status (late 20th century).

This study reveals that cohorts born at the end of the 19th century in Lanzarote were even somewhat taller than estimated averages for the Canary Islands as a whole. In the mid-1880s, the average height in the Canary Islands was 165.6 cm [32], while that of SB was 169.6 cm. If we compare the average of Mediterranean Spain—currently, the most robust study on heights of Spanish populations—at the end of the 19th century, when the height deteriorates or stagnates almost everywhere and decreases in the case studied, the average of SB was relatively taller (166.7 cm), higher than that of the average Spaniard who was 163.6 cm [64].

The height differences between Canarian and Mediterranean men decreased with island emigration, a phenomenon that the Canary Islands suffered from the beginning of the 20th century. The difference became only 0.9 cm in the 1905–1909 cohorts. During the 1920s the difference was 1.5 cm and in the autarchy decade (1940–1949) it was 2.2 cm. In the 1960s, in the midst of economic modernization, the average height in SB was 1.4 cm higher than that of the Mediterranean man, who on average was 171.8 cm. At the end of the period, we can compare with the Spanish average provided by the INE. In the 1973–1982 cohorts, SB averaged 177.7 cm, with a difference greater than that of Spain of 3.4 cm. The data suggest that the gains were important for the *lanzaroteños* or *conejeros.* The divergence with the height of the Spanish widened in the second half of the 20th century. If we compare the male height in SB with the data available for the Canary Islands as a whole, we verify that the balance was also favorable to the analyzed population. Thus, the difference in the 1934 cohorts (1955 recruitment) was 1.3 cm, 2.9 cm in 1941, 1 cm in 1961 and 0.8 cm in 1981 (2000 recruitment). The Canary Islands were then in the first place in the Spanish regional ranking together with the Basque Country [42].

As seen in other studies on the evolution of adult height, the main indicator of biological living standards [8,10,68,69,70], the SB data show a close relationship with the economic cycles of the island. At the end of the 19th century, the insular advantage was probably due to the improvements in foreign trade driven by the establishment of free trade. However, the turn of the century economic crisis impaired biological well-being. Emigration to Cuba and Venezuela deepened this deterioration in the first decades of the 20th century. Then, the divergence of biological well-being decreased compared to the rest (Canary Islands and Spain).

The situation has improved since the 1940 cohorts. The height growth trend reached its greatest upward inflection from the 1950s, in line with the strong economic growth in the “Spanish economic miracle” stage [69]. The traditional economy, such as the fishing activity in the Saharan fishing grounds and the canning fishing industry, registered a strong impulse after their capitalization and a deep technological transformation in their catches and commercial transformation, generating employment and income gains [64,65]. However, the development of the tourist industry was the economic engine of the islands. In this context, important improvements in transport, communications and drinking water infrastructures stand out [47]. The change in the economic model could have been the basis for the strong increase in height recorded between the 1940 and 1975 cohorts (Figure 5). The mean height increased 9.5 cm when it went from 169.9 to 179.4 cm. This was a large increase when compared to increases reported in other Spanish and European anthropometric studies [59,66,67].

The environmental transformations and the economic growth occurred jointly with demographic and epidemiological changes that modified the mortality and fecundity patterns. For the discussion, it is interesting to focus on health indicators, such as mortality. Around 1901–1905, the average infant mortality rate in the Canary Islands was very high, 180.6 infants dead per 1000 that were born and, in the 1940–1945 period, it was 70.1 per 1000, having diminished more than a half [71]. Since the 1930s the sequence of the demographic and epidemiological transition of the islands has been well documented. According to INE data, infant mortality has decreased considerably since then. In 1940, 109.8 infants per 1000 born died before their first birthday in the eastern Canary Islands, the province of Las Palmas de Gran Canaria, and this rate was almost half in 1950, at 61.7 per 1000, and, in 1970, it was 30.4 per 1000. In 1991, the general gross mortality rate in Lanzarote was 14.6 per 1000 in 1940 and decreased to 7.2 per 1000 in 1956–1960, under the Spanish average that was 9.1 per 1000. In 1981–1985 it was 5.6 per 1000 against the Spanish rate of 7.7 per 1000 [53]. The environmental changes produced since the 1940s, without neglecting the effect of the islands’ benign climate may have favored islander health in this period since the middle of the 20th century after the diffusion of antibiotics (penicillin), sulfonamides and compulsory vaccination in the 1960s. The infant mortality rates in 1940 and 1950 in the province were also slightly lower than the average in Spain.

Numerous studies show the long-term adverse effects of early exposure to a wide range of infectious diseases on health and socioeconomic outcomes in adulthood. In the past, infectious diseases related to poor water infrastructure and poor food conditions were responsible for high infant mortality mainly associated with diseases of the digestive system. Early exposure to infectious diseases and epidemics caused stunting. The intensity of exposure had an effect on the prevalence of stunting [72,73,74]. A statistical method to evaluate the growth and nutritional status of infants and adolescents with anthropometric parameters, such as height, weight and BMI, is the z-score [75]. Disseminated by pediatricians and the WHO, z-scores have been used by historians and specialists to assess malnutrition and stunting in historical populations [24,76,77,78]. In the population analyzed, stunting is irrelevant.

Another health indicator that, like infant mortality, is inversely related to height is life expectancy [28,79]. In the eastern Canaries, life expectancy at birth was 51.2 years in 1940, while in Spain it was 50.1 years. Thirty years later, in 1970, it stood at 71.9 and 72.4 years, respectively. Spanish populations improved their standards of living and health as measured by this indicator during the years of outstanding economic growth. For later dates, the dynamics of life expectancy at birth in men and women are better documented. For the former, it went from 69.9 years in 1972 to 74.7 years in 2000, somewhat lower than the Spanish averages, 70 and 75.6 years, respectively [80]. However, in any case, the Spanish are among the people with highest gains in longevity [81]. The combination of social, economic, sanitary and epidemiological improvements explains the evolution of life expectancy and they are also behind the increase in height. It would be interesting to explore the role of these two indicators with more precise data from the island and the population analyzed to explore the relationships established between health and anthropometrics. We cannot rule out that the relative insular geographic isolation also became a relative advantage in the face of epidemics that frequently sowed disease and death in the peninsular populations. However, this matter requires further study.

Diet also contributed to the improvement of their nutritional status. The data show that the height analyzed in Lanzarote was as high or higher than the averages of the Canary Island inhabitants as a whole throughout the 20th century. Likewise, we know that the Canaries were, together with the population of the Basque Country, the tallest populations in Spain, where the consumption of meat and especially milk was decisive at the beginning of the 20th century [82]. An abundant body of literature shows that the consumption of animal proteins, mainly milk and meat, determines the nutritional status in the two critical stages of child growth: childhood and adolescence [83,84]. Milk is a complete and well-balanced source of the nutrients and energy necessary to ensure child growth and development [85]. Evidence suggests the positive effects of dairy products and particularly milk proteins on post-neonatal linear growth even in adolescents, who could regain some of the growth lost due to malnutrition in critical phases of childhood [86,87].

We do not have precise data on the evolution of animal protein consumption for the case analyzed, not even for the island or the whole of the Canary Islands, but we do have some data that confirm the high consumption of calcium derived from milk among the Canary Island inhabitants. In addition to high levels of energy, protein and micro- and macronutrients, milk contains calcium and insulin-like growth factor 1, which are of great importance for the development and growth of children [87]. Recent studies on the evolution of dairy consumption in Spain show its growing importance from the first decades of the 20th century and especially since 1960 [82,88,89,90]. In Lanzarote and other Canary Islands, there were advances in sheep and goat farming, as well as pigs, and the consumption of milk from goats of the *majorera* breed has been well documented since the 15th century [91]. In the mid-1960s, an investigation carried out on nutrient needs in Spain revealed that the province of Las Palmas de Gran Canaria, which includes the island of Lanzarote, stands out for its high consumption of milk calcium with the highest proportion in Spain, including the western Canary Islands [92].

In addition to calcium from dairy products, one must consider fish consumption which has traditionally been high in the region due to its favorable local income. Finally, a nutritional singularity of the island of Lanzarote is *gofio*, a product made from ground flours of various toasted cereals, generally wheat and corn or millet, introduced from America. This food preparation carried out mainly in flour windmills, very typical milling devices of the island of Lanzarote (Figure 14), spread in the 19th and early 20th centuries, also using barley, wheat and even lentils. Due to its high nutritional content, it became the basic food of the Canary Island peasant inhabitants and was essential to alleviate the times of famine that the archipelago suffered in successive periods. SB became one of the great gofio-producing centers from the mid-19th century [93].

## 6. Conclusions

This paper’s findings can be summarized as follows. Firstly, the height trend is in line with what we know about average heights for the Canarian Island inhabitants, who have historically been taller than the average in the Spanish population in the 19th and 20th century. The data suggest that the biological well-being of the Canary Island inhabitants was slightly healthier than those of the Iberian Peninsula. Secondly, the evolution of height was not linear for all social groups. Inequality persists at the beginning and the end of the period, and results based on coefficients of variation (CVs) and percentiles even show an increase in inequality levels from the 1960s until the end of the 20th century.

This article provides knowledge on the evolution of the physical well-being of the Spanish Atlantic populations. Based on a case study of the eastern Canary Islands, the findings show the impact of socioeconomic processes on the biological standard of living and inequality. They also show the importance of the relationships between physical growth and economic growth, such as the nutritional and epidemiological transition. Among the main determinants of the advantage of the Canary Island height with respect to that of the Iberian Peninsula, the uniqueness of the diet and the improvement of environmental conditions stand out. The findings are important for the implications for economic history, but also for physical anthropology that analyzes plasticity and biological variability.

We can state that the definitive prevention of any manifestation of malnutrition (serious or chronic) might be due the improvement of the diet and hygiene and the distribution of drinking water, among others factors, since the beginning of the 1960s and also because of the increases in BMI.

## Figures and Tables

**Figure 1 ijerph-18-12843-f001:**
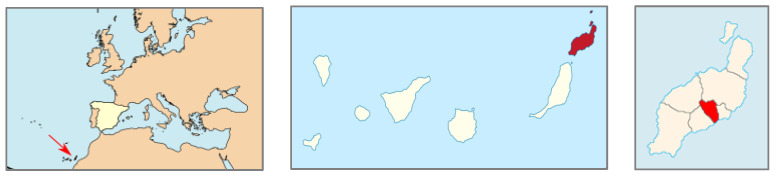
(**Left**): Location of the Canary Islands, a region of Spain. (**Center**): Canary Islands and Lanzarote in red. (**Right**): Lanzarote and SB town in red.

**Figure 2 ijerph-18-12843-f002:**
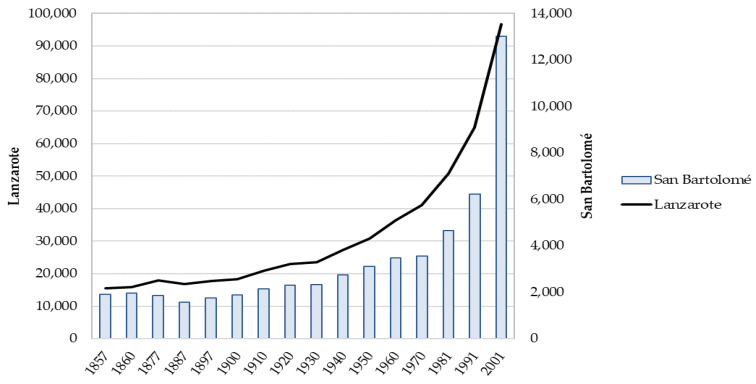
Evolution of the population in the municipality of SB and Lanzarote, 1857–2020.Source: own work from INE data.

**Figure 3 ijerph-18-12843-f003:**
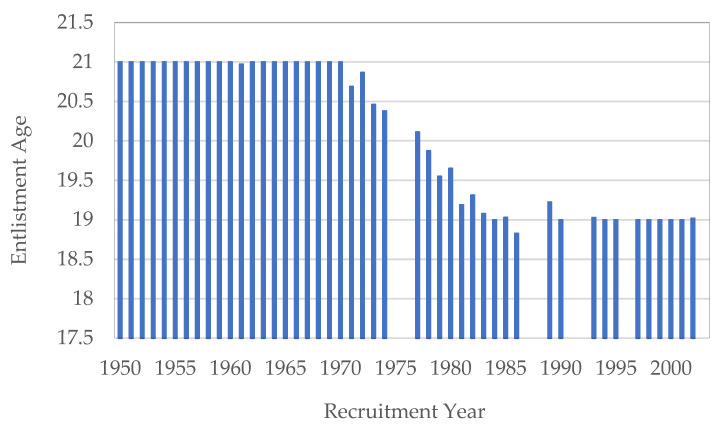
Changes in the recruitment regulatory age between 1969 and 1986. Source: AMSB, ACDS. Own work.

**Figure 4 ijerph-18-12843-f004:**
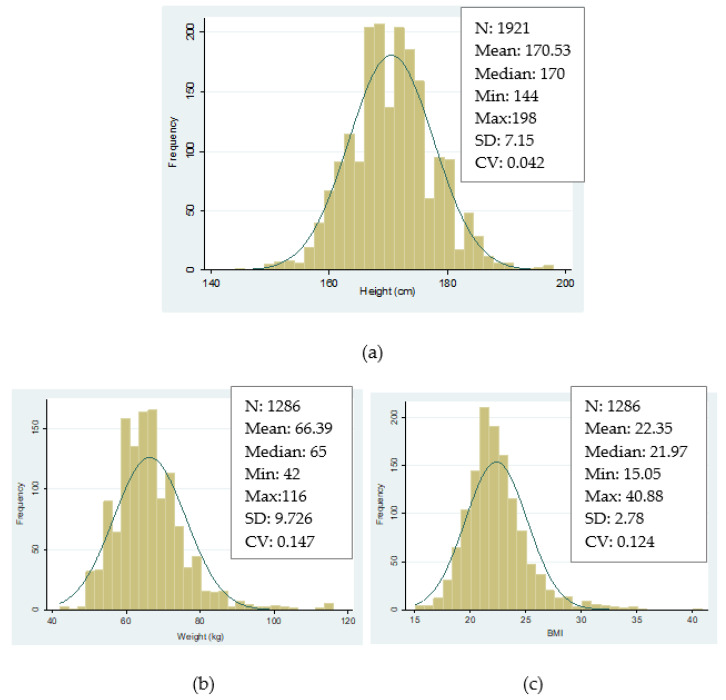
Histograms. (**a**) Height distributions in SB, birth cohorts 1886–1982. (**b**) Weight distributions in SB, birth cohorts 1886–1982. (**c**) Body mass index in SB, birth cohorts 1886–1982. Source: AMSB, ACDS. Own work.

**Figure 5 ijerph-18-12843-f005:**
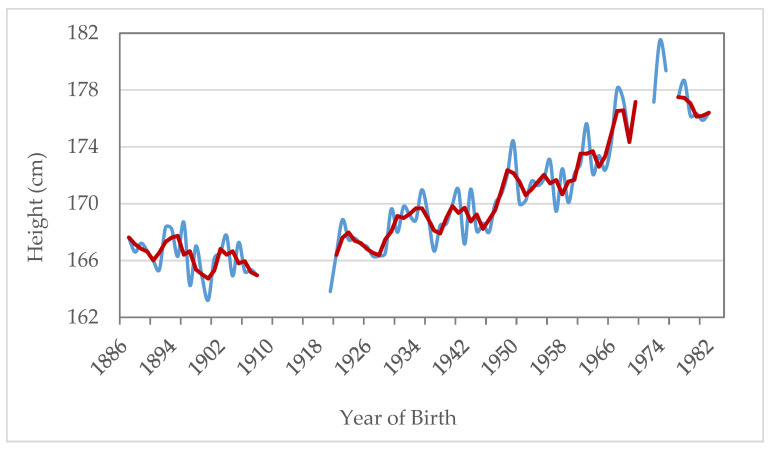
Annual average heights in SB. Cohorts 1886–1982. Estimate with 3rd order moving averages. (MM3). Source: AMSB, ACDS. Own work.

**Figure 6 ijerph-18-12843-f006:**
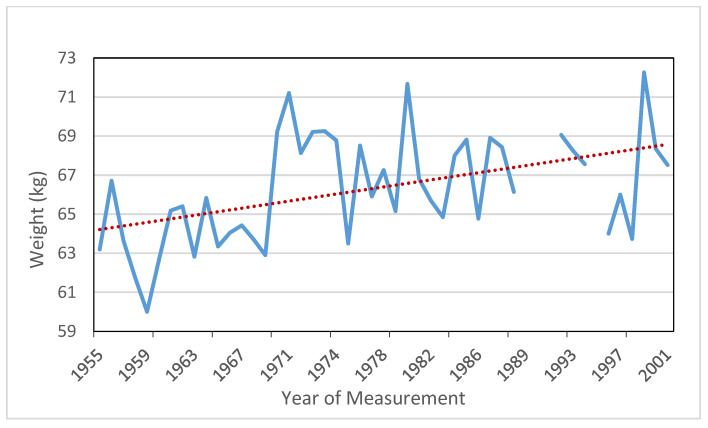
Annual average weights (1955–2001) in SB. Trend line. Source: AMSB, ACDS. Own work.

**Figure 7 ijerph-18-12843-f007:**
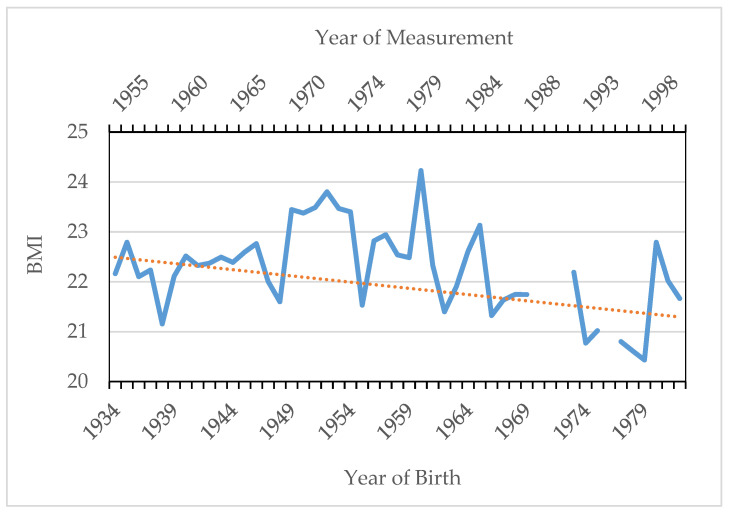
Annual average BMI (1955–2001) in SB. Trend line. Source: AMSB, ACDS. Own work.

**Figure 8 ijerph-18-12843-f008:**
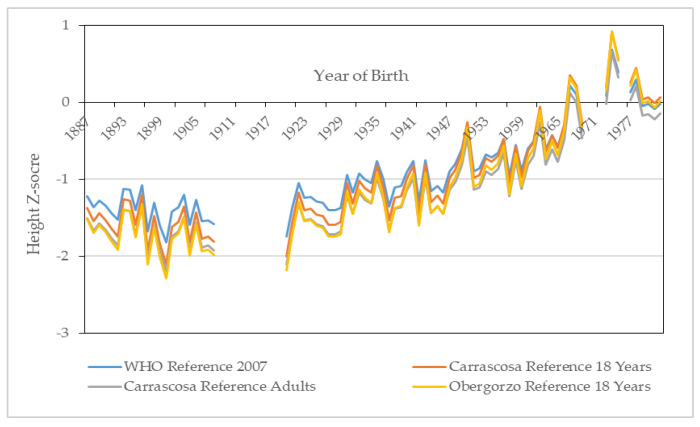
Annual height z-scores (HAZ). Source: AMSB, ACDS. Own work.

**Figure 9 ijerph-18-12843-f009:**
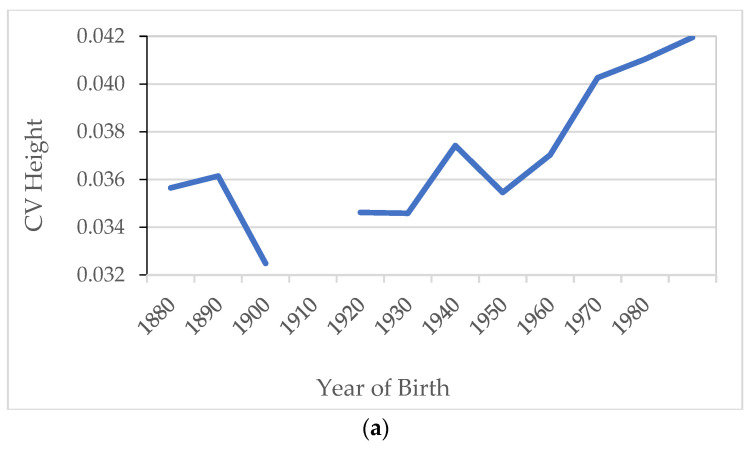
(**a**) Height coefficient of variation by decade of birth. (**b**) BMI coefficient of variation by decade of birth. Source: AMSB, ACDS. Own work.

**Figure 10 ijerph-18-12843-f010:**
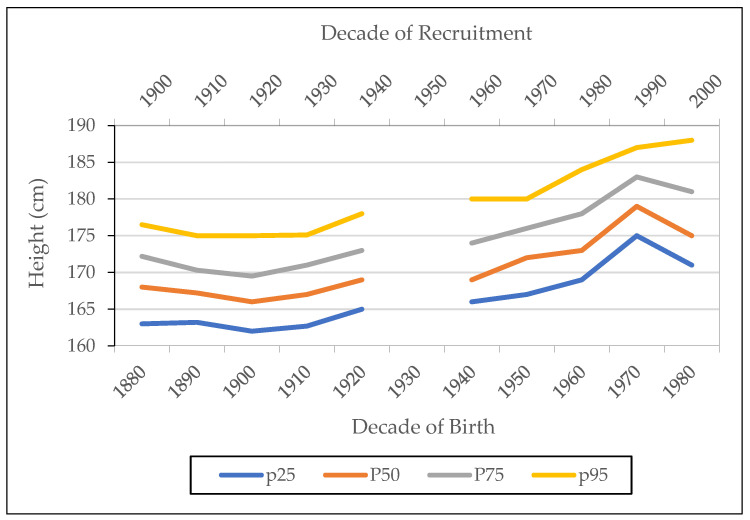
Height percentiles by decade of birth/recruitment. Source: AMSB, ACDS. Own work.

**Figure 11 ijerph-18-12843-f011:**
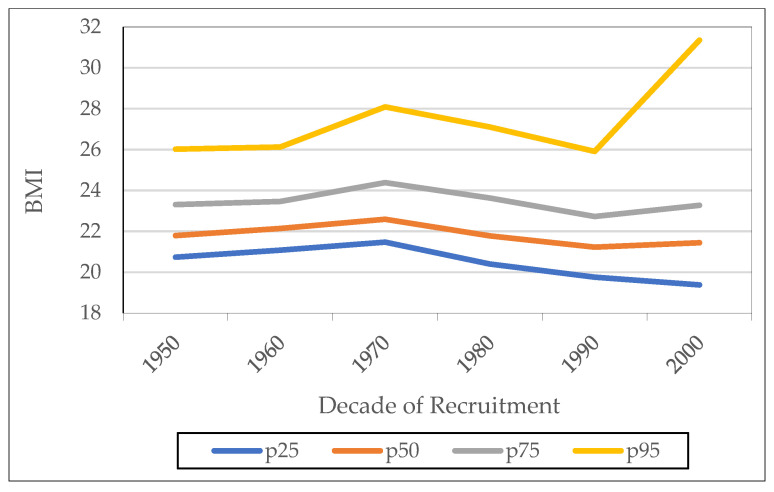
BMI percentiles by decade of recruitment Source: AMSB, ACDS. Own work.

**Figure 12 ijerph-18-12843-f012:**
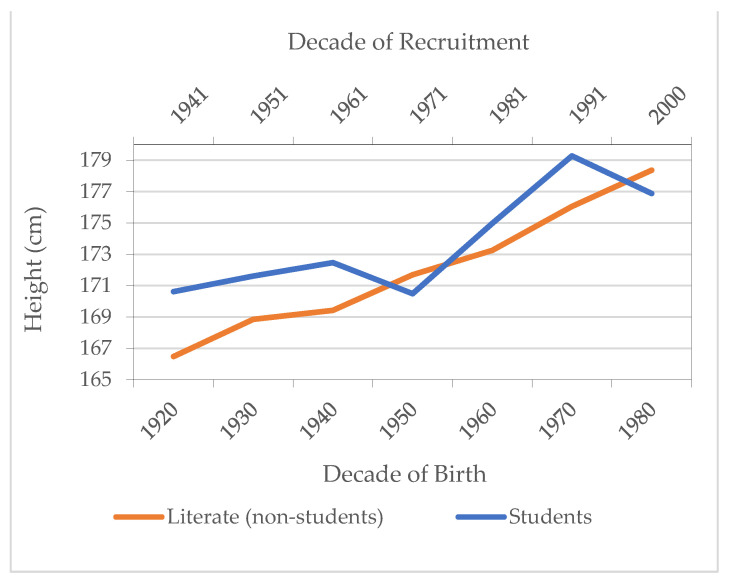
Average height by literate non-students and students by year of birth/recruitment. Source: AMSB, ACDS. Own work.

**Figure 13 ijerph-18-12843-f013:**
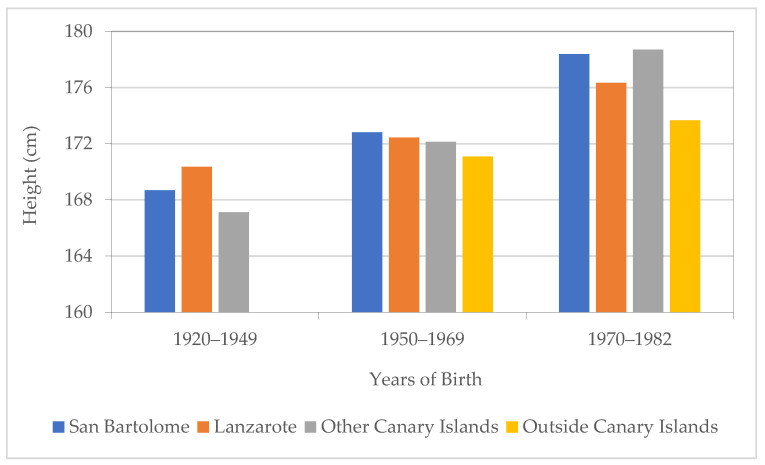
Average height by origin and year of birth. Source: AMSB, ACDS. Own work.

**Figure 14 ijerph-18-12843-f014:**
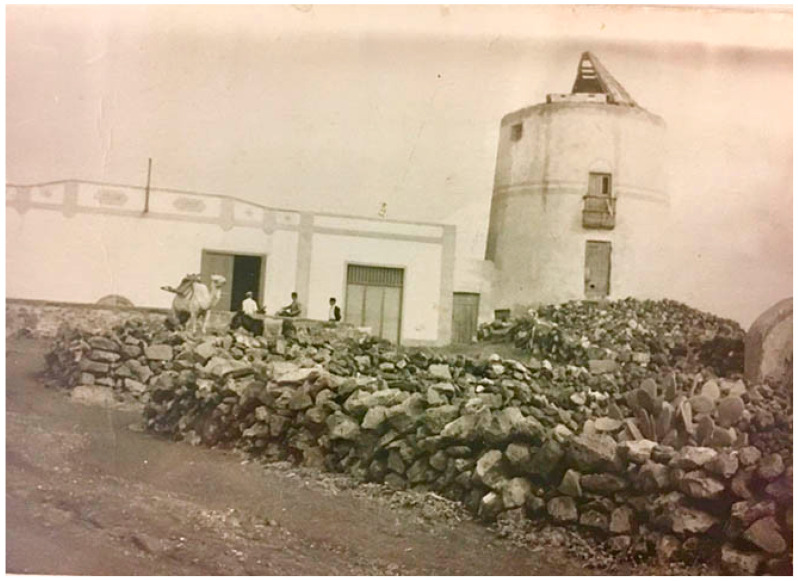
Flour mill of SB (1870), erected by Baltasar Fermín and acquired by José María Gil in 1919, today declared a Site of Cultural Interest (BIC).

**Table 1 ijerph-18-12843-t001:** Sample size by variable of interest.

Variable of Interest	Observations
Number of young men called up	2172
Height	1921
Missing height information	251
Weight and BMI	1286
Missing weight and BMI information	886
Literacy	1941
Missing literacy information	231
Occupation	1523
Missing occupation information	649

Source: AMSB, ACDS. Own work.

**Table 2 ijerph-18-12843-t002:** Descriptive statistics by decade of birth/recruitment.

		Height (cm)				Weight (kg)				IMC			
Decade of Birth	Decade of Recruitment	N	Mean	SD	CV	N	Mean	SD	CV	N	Mean	SD	CV
1886–1889	1907–1910	65	167.08	5.957	0.036	1	57.00			1	24.38		
1890–1899	1911–1920	153	166.59	6.022	0.036	22	62.32	6.578	0.106	22	22.74	1.690	0.074
1900–1909	1921–1930	175	165.75	5.385	0.032								
1910–1919	1931–1940	1	163.10										
1920–1929	1941–1950	169	166.87	5.778	0.035								
1930–1939	1951–1960	229	169.01	5.845	0.035	140	63.39	7.505	0.118	140	22.21	2.194	0.099
1940–1949	1961–1970	272	169.54	6.344	0.037	269	64.64	7.982	0.123	269	22.45	2.211	0.098
1950–1959	1971–1980	241	171.19	6.070	0.035	241	67.73	9.608	0.142	241	23.06	2.570	0.111
1960–1969	1981–1990	360	173.74	6.434	0.037	360	66.87	9.217	0.138	360	22.15	2.811	0.127
1970–1979	1991–1999	83	178.06	7.169	0.040	81	67.68	9.428	0.139	81	21.38	2.535	0.119
1980–1982	2000–2001	173	176.14	7.228	0.041	172	68.69	13.655	0.199	172	22.09	3.976	0.180
Total		1921	170.53	7.154	0.042	1286	66.39	9.726	0.146	1286	22.35	2.778	0.124

Source: AMSB, ACDS. Own work.

**Table 3 ijerph-18-12843-t003:** Statistical information by BMI classification and decade of birth/recruitment.

Birth Decade	Recruitment Decade	Total	Low Weight					Normal Weight					Overweight or Obesity				
		N	N	Mean	SD	CI 95%		N	Mean	SD	CI 95%		N	Mean	SD	CI 95%	
1930s	1950s	140	1	51				126	62	0.49	61	63	13	77.8	2.44	73	83
1940s	1960s	269	5	54.6	1.96	49	60	239	63.4	0.41	63	64	25	78.4	1.81	75	82
1950s	1970s	241	2	54.5	2.5	23	86	189	64.5	0.46	64	65	50	80.3	1.34	78	83
1960s	1980s	360	26	54.7	0.96	53	57	294	65.8	0.37	65	67	40	82.8	1.58	80	86
1970s	1990s	81	7	55	2.18	50	60	69	67.8	0.84	66	69	5	83.4	7.95	61	105
1980s	2000s	172	19	54.3	1.34	51	57	126	65.8	0.64	65	67	27	92.4	2.58	87	98
Total		1263	60					1043					160				

Source: AMSB, ACDS. Own work.

**Table 4 ijerph-18-12843-t004:** Stunting rates according to the WHO (2007) by decade of birth/recruitment.

Decade of Birth	Decade of Recruitment	N	Standard HAZ < −1 Cases	%	Standard HAZ < −2 Cases	%
1880s	1900s	64	64	100		
1890s	1910s	153	153	100		
1900s	1920s	175	175	100		
1910s	1930s					
1920s	1940s	169	169	100		
1930s	1950s	229	229	100		
1940s	1960s	272	113	41.54		
1950s	1970s	241				
1960s	1980s	360				
1970s	1990s	83				
1980s	2000s	173				
		1919	903	47.06	0	0

Source: AMSB, ACDS. Own work. No data for 1930 recruitments due to emigration and crisis.

**Table 5 ijerph-18-12843-t005:** Summary statistics by social class group (full sample).

	Height (cm)					BMI				
	n	Mean	SD	Min	Max	n	Mean	SD	Min	Max
Students	318	175.32	7.11	158	197	303	22.26	3.30	16	41
Non-manual workers	174	173.36	6.95	156	190	165	22.22	2.69	17	32
Manual workers	350	171.68	5.99	152	198	332	22.50	2.62	15	33
Farmers	220	168.63	6.18	149	184	132	22.62	2.63	18	35
Farm workers	389	168.59	6.18	144	190	226	22.37	1.90	18	31
Total	1451	171.39	6.97	144	198	1158	22.38	2.70	15	41

Source: AMSB, ACDS. Own work.

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
