# Peer review of "Biological Well-Being and Inequality in Canary Islands: Lanzarote (Cohorts 1886–1982)"

_ijerph, 2021, doi:10.3390/ijerph182312843_

Round 1

Reviewer 1 Report

The manuscript addresses a topic of interest, the analysis carried out in the Spanish anthropometric context yields relevant information that may have an impact on the design of new public policies.

The methodological section is a bit confusing and may hinder the replicability of the study. I recommend the authors to re-elaborate this section in a logical and coherent way. In the presentation of the results some figures present obsolete data, it would be good if the authors update the data.  The authors should pay attention to the temporalities used in the study, there is a lot of heterogeneity and this detracts from the quality of the study. 

In the discussion section it seems that the conclusions of the study are presented or the results are repeated, there is no clarity in the discussion of the results and confrontation with other studies on the topic addressed, the theoretical basis of the study should be present in this section. The conclusions section is deficient, the abundant information obtained is not taken advantage of. The bibliography used is deficient and tends towards obsolescence; it is important that the authors expand and update the bibliography of the study.

In general the study presented by the authors is interesting, the improvements suggested previously could give it a better presentation and put it in a better position for publication.

Reviewer 2 Report

I attach my considerations

Reviewer 3 Report

Authors have appropriately described their approach to analysis this historic data. However, following minor suggestions will improve the quality of reporting and interpretability of methods and results.

Adding statistical significance to some results will be worthwhile as it will add an additional value to those results. For example, an increasing trend for weight (Kg) was presented in the figure 6, and by adding a p-value will tell that this was significantly increased over time. Similarly, some other results may be considered especially where some comparison has been done.

Table 3 showed the number of observations and percentage by BMI classification and decade of birth/recruitment. It will have more worth if 95% confidence interval as it will tell uncertainty bounds around the estimate. It is also important to remember that interval estimates are better than point estimates. Similarly, some other results may be considered as appropriate.
